# Differential virulence of *Trypanosoma brucei rhodesiense* isolates does not influence the outcome of treatment with anti-trypanosomal drugs in the mouse model

Kariuki Ndung'u[1]*, Grace Adira Murilla[1,2], John Kibuthu Thuita[1,3], Geoffrey Njuguna Ngae[4], Joanna Eseri Auma[1], Purity Kaari Gitonga[1], Daniel Kahiga Thungu[1], Richard Kiptum Kurgat[1], Judith Kusimba Chemuliti[1], Raymond Ellie Mdachi[1]

1 Biotechnology Research Institute, Kenya Agricultural and Livestock Research Organization, Kikuyu, Kenya, 2 KAG EAST University, Nairobi, Kenya, 3 Meru University of Science and Technology, Meru, Kenya, 4 Food Crops Research Institute, Kenya Agricultural and Livestock Research Organization, Nairobi, Kenya

* Kariukindungu1960@gmail.com

**Data Availability Statement:** All relevant data are within the manuscript and its Supporting Information files.

## Abstract

We assessed the virulence and anti-trypanosomal drug sensitivity patterns of *Trypanosoma brucei rhodesiense* (*Tbr*) isolates in the Kenya Agricultural and Livestock Research Organization-Biotechnology Research Institute (KALRO-BioRI) cryobank. Specifically, the study focused on *Tbr* clones originally isolated from the western Kenya/eastern Uganda focus of human African Trypanosomiasis (HAT). Twelve (12) *Tbr* clones were assessed for virulence using groups(n = 10) of Swiss White Mice monitored for 60 days post infection (dpi). Based on survival time, four classes of virulence were identified: (a) very-acute: 0–15, (b) acute: 16–30, (c) sub-acute: 31–45 and (d) chronic: 46–60 dpi. Other virulence biomarkers identified included: pre-patent period (pp), parasitaemia progression, packed cell volume (PCV) and body weight changes. The test *Tbr* clones together with KALRO-BioRi reference drug-resistant and drug sensitive isolates were then tested for sensitivity to melarsoprol (mel B), pentamidine, diminazene aceturate and suramin, using mice groups (n = 5) treated with single doses of each drug at 24 hours post infection. Our results showed that the clones were distributed among four classes of virulence as follows: 3/12 (very-acute), 3/12 (acute), 2/12 (sub-acute) and 4/12 (chronic) isolates. Differences in survivorship, parasitaemia progression and PCV were significant (P<0.001) and correlated. The isolate considered to be drug resistant at KALRO-BioRI, KETRI 2538, was confirmed to be resistant to melarsoprol, pentamidine and diminazene aceturate but it was not resistant to suramin. A cure rate of at least 80% was achieved for all test isolates with melarsoprol (1mg/Kg and 20 mg/kg), pentamidine (5 and 20 mg/kg), diminazene aceturate (5 mg/kg) and suramin (5 mg/kg) indicating that the isolates were not resistant to any of the drugs despite the differences in virulence. This study provides evidence of variations in virulence of *Tbr* clones from a single HAT focus and confirms that this variations is not a significant determinant of isolate sensitivity to anti-trypanosomal drugs.

**Funding:** Materials were provided by the Kenya Government. The funder had no role in study design, data collection and analysis, decision to publish, or preparation of the manuscript.

**Competing interests:** The authors have declared that no competing interests exist.

## Introduction

Human African trypanosomiasis (HAT), also known as sleeping sickness, is a vector-borne parasitic disease. It is caused by infection of humans with protozoan parasites belonging to the genus *Trypanosoma*. HAT is caused by two subspecies of trypanosomes, namely *Trypanosoma brucei gambiense* and *Trypanosoma brucei rhodesiense* [1]. They are transmitted to humans by tsetse flies (*Glossina* genus) [2]. *Trypanosoma brucei gambiense* is found in countries in West and Central Africa and causes a chronic infection [3]. A person can be infected for months or even years without major signs or symptoms of the disease [4]. When more evident symptoms emerge, the patient is often already in an advanced disease stage where the central nervous system is affected. *Trypanosoma brucei rhodesiense* is found in countries in eastern and southern Africa and causes an acute infection. Symptoms manifest within 2–4 weeks of an infective tsetse fly bite [3]. HAT develops in two stages namely, the early (hemolymphatic) and the late (meningo-encephalitic) stage. In the early stage of the disease, parasites proliferate in the blood and lymphatic system while in the late stage, parasites penetrate the blood brain barrier (BBB) and persist and proliferate in the central nervous system (CNS), causing an encephalitic reaction that leads to death if untreated or inadequately treated [5]. For first stage infections, there are no specific clinical signs and symptoms in both forms of the disease; fever, headache and loss of appetite are common [1] as well as anaemia in the monkey model [6]. With *T.b. rhodesiense infections*, first signs and symptoms are observed a few weeks after infection [1]. However, a mild form of chronic *T. b. rhodesiense* infections with incubation times of several months has been reported in Zambia [7]. The acute and the chronic HAT infections caused by *T. b. rhodesiense* in different foci differ in both their inflammatory response and pathology. According to Maclean et al (2008), the pathology encountered in the acute HAT infections is characterized by elevated Tumor necrosis factor alpha (TNF-α) while that encountered in the chronic HAT infections is characterized by elevated transforming growth factor (TGF-β) [8].

Treatment of *Trypanosoma brucei rhodesiense* infections involves the use of early stage drugs such as pentamidine and suramin [9] and late stage drugs such as melarsoprol; melarsoprol is the only drug recommended by WHO for treatment of late-stage *T b rhodesiense* infection, but can be lethal to 5% of patients owing to post-treatment reactive encephalopathy [10]. HAT therapy is further complicated by reports of drug resistance in different foci, including against suramin and melarsoprol in Tanzania [11] and against melarsoprol in south Sudan [12]. In their study, Pyana and colleagues [13] suggested that investigations into treatment failure in HAT and use of alternative drugs or treatment regimens should not only focus on differential genotypes of the parasites but also on differential virulence and tissue tropism as possible causes. The present study was therefore designed to investigate the occurrence of differential virulence of isolates recovered from Western Kenya/ Eastern Uganda HAT focus and the potential role of this variations on isolate sensitivity to anti-trypanosomal drugs using the mouse model. Studies on disease pathogenesis, parasite virulence, drug sensitivity and identification of new potential drug targets and staging biomarkers are commonly carried out in the mouse model based on its cost effectiveness, genetic similarity to humans estimated to be 85%, and ethical limitations of carrying out such studies in higher animal models or humans [14–16]. The study will also avail well characterized *T.b.rhodesiense* isolates for future studies.

## Materials and methods

### Ethics

All experimental protocols and procedures used in this study involving laboratory animals were reviewed and approved by Institutional Animal Care and Use Committee (IACUC) of

Kenya Agricultural and Livestock Research Institute–Biotechnology Research Institute (KAL-RO-BioRI) Ref: C/Biori/4/325/II/53)

## Experimental animals

The study used 6to 8 weeks old male Swiss White mice, each weighing 25–30 g live body weight. The animals were obtained from the Animal Breeding Unit at KALRO-BioRI, Muguga. The mice were housed in standard mouse cages and maintained on a diet consisting of commercial pellets (Unga® Kenya Ltd). All experiments were performed according to the guidelines set by the Institutional Animal Care and Use Committee (IACUC) of KALRO--BioRI. Briefly, water was provided ad libitum. All mice were acclimatized for two weeks, during which time they were screened and treated for ecto and endoparasites using ivermectin (Ivermectin®, Anupco, Suffolk, England). During the two-week acclimation period, pre-infection data were collected on body weights and packed cell volume once a week prior to parasite inoculation.

## Trypanosomes and cloning

Twelve *T.b. rhodesiense* trypanosome stabilates (KETRI 2482, 2487, 3304, 3305, 3380, 3664, 3798, 3800, 3801, 3803, 3926, 3928) were selected from the KALRO-BioRI specimen bank. Cloning was carried out as described by [17]. Briefly, the trypanosome stabilates were inoculated into mice immunosuppressed using cyclophosphamide at 100 mg/kg for three consecutive days (total dose 300mg/kg) body weight (bwt) as previously described [18]. Animals were monitored daily for parasitaemia. When the mice attained a parasitaemia score of $3.2.x10^7$ trypanosomes/mL [19], they were bled from the tail vein and the blood sample appropriately diluted using a mixture of PSG pH 8.0 and guinea pig serum in the ratio of 1:1. Using the hanging drop method [20], a single trypanosome was then picked using a syringe with a 25 gauge needle suspended in at least 0.2mls PSG pH 8.0 and injected intraperitoneally (ip) into a single immunosuppressed mouse. This was replicated ten times to increase the chances of success. Infected mice were then monitored for parasitaemia daily [19]. Any of the ten mice which became parasitaemic was euthanized using concentrated carbon dioxide, bled from the heart and the harvested trypanosomes cryopreserved in PSG pH 8.0 in 20% glycerol as a clone stabilate.

## Virulence studies

**Design of virulence study.**   Male Swiss White mice were housed in groups of 10 in standard mouse cages containing wood shavings as bedding material. The cryopreserved cloned parasites were thawed, and injected ip into immunosuppressed donor Swiss White mice for multiplication. The mice were euthanized using carbon dioxide [21] at peak parasitaemia and blood collected from the heart in EDTA for quantification as previously described [22]. The ten mice in each cage were infected with one *Tbr* clone, with each mouse receiving $1x10^4$ trypanosomes injected intraperitoneally. The infected mice were monitored for pre-patent period, parasitaemia progression, PCV, body weight and survival time as virulence biomarkers. A control group of 10 non-infected mice was included in the study and monitored for changes in PCV, body weight and survival times.

**Pre-patent period and parasitaemia progression.**   Blood for estimation of parasitaemia levels was collected daily for the first 14 days and thereafter three times in a week from each mouse using the tail tip amputation method [23]. The PP and parasitaemia progression were determined using the rapid matching method of [19, 24]. The infected mice were monitored for 60 days post infection.

**Table 1. Changes in virulence biomarkers in mice infected with twelve clones of *Trypanosoma brucei rhodesiense*.**

| Class | Clone ID | Locality of isolation | Iso. yr | PP passage No | Peak Para. | DPP | MST |
|---|---|---|---|---|---|---|---|
| very-acute | KETRI 2482 | Lumino, Uganda | 1969 | 5±0 7 | $1x10^6$ | 8±0.2 | 9±0.4 |
| | KETRI 3304 | Lugala, Uganda | 1971 | 5±0 64 | $7.6x10^8$ | 7.8±0.4 | 9±0.4 |
| | KETRI 3803 | Busia, Kenya | 1961 | 4±0 2 | $9.3x10^8$ | 6.4±0.2 | 8.2±0.3 |
| | *Group mean ±SE* | | | *4.7±0.9* | *$8.9x10^8$* | *7.4±0.5* | *8.8±0.2* |
| Acute | KETRI 2487 | Busoga, Uganda | 1972 | 4±0 5 | $6.2x10^8$ | 7±0 | 18.1±*0.7* |
| | KETRI 3800 | Busia, Kenya | 2000 | 5.2±0.1 2 | $1.2x10^8$ | 6±0 | 26.0±*2.0* |
| | KETRI 3801 | Busia, Kenya | 1989 | 6±0 1 | $6.8x10^7$ | 7.3±0.3 | 20.4±*0.8* |
| | *Group mean ±SE* | | | *5.03±0.16* | *$1.7x10^8$* | *6.7±1.3* | *21.6±1.0* |
| Sub-acute | KETRI 3798 | Busia, Kenya | 1989 | 5.3±0.16 2 | $1.6x10^8$ | 9.3±1.7 | 28.2±*2.0* |
| | KETRI 3926 | Busoga, Uganda | 1972 | 5±0 8 | $1.7x10^8$ | 6.6±0.2 | 38.9±*1.6* |
| | *Group mean ±SE* | | | *5.2±0.8* | *$1.5x10^8$* | *7.9±0.9* | *33.6±1.8* |
| Chronic | KETRI 3928 | Tororo, Uganda | 1992 | 6.3±0.3 2 | $7.9x10^7$ | 7.0±0 | 51.8±*4.1* |
| | KETRI 3664 | Busia, Kenya | 1997 | 6.0±0 2 | $4.8x10^7$ | 7.2±0.3 | 45.6±*1.6* |
| | KETRI 3380 | Busoga, Uganda | 2000 | 5.9±0.5 3 | $1.3x10^8$ | 7.3±0.2 | 55.5±*2.3* |
| | KETRI 3305 | Lugala, Uganda | 1971 | 6.7±0.2 5 | $3.2x10^8$ | 8.7±0.3 | 46.7±*5.1* |
| | *Group mean ±SE* | | | *6.3±0.2* | *$1.1x10^8$* | *7.5±0.2* | *49.9±1.8* |

Control Non-infected    -    -    -    -    -    >60

Key: PP-pre-patent period, Iso Yr–year of isolation, Par-parasitaemia, DPP -days to peak parasitaemia, MST-mean survival times,—No data

**Packed cell volume (PCV) and body weight changes.** PCV was determined as outlined by [25]. Body weight (bwt) was measured once a week using a (Mettler Tolendo PB 302 ®, Switzerland) digital balance [22].

**Survival times and virulence classification.** The classification of trypanosome virulence was based on the survival of the infected mice as previously described [26]. The twelve *T.b. rhodesiense* clones were placed into four classes of virulence based on the survival of 60% or more of the infected mice as follows: very-acute (0–15 days), acute (16–30), sub-acute (31–45) and chronic classes (46–60). Each mouse's survival time was determined on the basis of a $\geq 25\%$ decline in PCV and consistently high parasitaemia levels of $1x10^9$/ml for at least two consecutive days as previously described [27]. The mice were immediately euthanized by CO2 asphyxiation following the Institutional Animal Care and Use Committee (IUCAC) guidelines as earlier described by [28] and recorded as dead animal. Mice surviving at 60 dpi were equally euthanized, survival time recorded as 60 days and categorized as censored data.

**Drug sensitivity study.** Initially, sensitivity patterns for KALRO-BiORI laboratory reference isolates considered drug resistant (KETRI 2538) or drug-sensitive (KETRI 3738) were determined for Melarsoprol (Arsobal®, Aventis), Diminazene aceturate [(Veriben®, Ceva, France), Pentamidine (Pentacarinat®-Sanofi, UK) and Suramin (Germanin® Bayer), using dose rates ranging from 1–40 mg/kg body weight (Table 2) in order to identify cut-off points for characterizing isolates as drug resistant. Thereafter, the *T. b. rhodesiense* test clones were evaluated for sensitivity to the same drugs (Table 3). An isolate was considered drug-resistant if 2/5 (40%) of the infected and treated mice relapsed [11] after having been treated at 20mg/kg bwt.

Suramin and Pentamidine drugs (100% w/w) for the highest dosage of 40mg/kg bw were prepared by dissolving 40mg of these drugs in 10mls distilled water to give a concentration of 4 mg/ml. Diminazene aceturate (44.44% w/w active ingredient) for the highest dosage of 40mg/kg bw was prepared by dissolving 90mg of the drug powder in 10 mLs distilled water to

**Table 2. Results of drug sensitivity evaluation of reference KALRO-BioRI sensitive and resistant *T b rhodesiense* isolates.**

| | *Sensitive Isolate KETRI 2537* | | | *Resistant isolate KETRI 2538* | | |
|---|---|---|---|---|---|---|
| Drug | Drug dose (mg/Kg) | Mice cured/5 | Status | Drug dose (mg/Kg) | Mice cured/5 | Status |
| MelB | 40 | 5 | (s) | 40 | **5** | **(S)** |
| | 20 | 4 | (s) | 20 | 0 | (R) |
| | 10 | 5 | (s) | 10 | 0 | (R) |
| | 5 | 5 | (s) | 5 | 0 | (R) |
| | 2.5 | 5 | (s) | 2.5 | 0 | (R) |
| | 1 | 0/5 | (R | 1 | 0 | (R) |
| diminazene aceturate | 40 | 5 | (s) | 40 | 5 | **(S)** |
| | 20 | 5 | (s) | 20 | 0 | (R) |
| | 10 | 4 | (s) | 10 | 0 | (R) |
| | 5 | 5 | (s) | 5 | 0 | (R) |
| | 2.5 | 5 | (s) | 2.5 | 0 | (R) |
| | 1 | 2 | **(R)** | 1 | 0 | (R) |
| Pentamindine | 40 | 5 | (s) | 40 | 5 | **(S)** |
| | 20 | 5 | (s) | 20 | 1 | (R) |
| | 10 | 5 | (s) | 10 | 1 | (R) |
| | 5 | 4 | (s) | 5 | 0 | (R) |
| | 2.5 | 2 | **(R)** | 2.5 | 0 | (R) |
| | 1 | 0 | **(R)** | 1 | 0 | (R) |
| Suramin Control | 40 | 5 | (S) | 40 | 5 | **(S)** |
| | 20 | 5 | (S) | 20 | 5 | **(S)** |
| | 10 | 5 | (s) | 10 | 5 | **(S)** |
| | 5 | 5 | (s) | 5 | 5 | **(S)** |
| | 2.5 | 4 | (s) | 2.5 | 2 | (R) |
| | 1 | 1 | **(R)** | 1 | 0 | (R) |
| | - | 10 | | - | 10 | |

Key:- Not treated; The mice groups (n = 5) were treated 24hours post inoculation with the isolates and monitored for 60 days post treatment. An isolate is coded as sensitive (S) when at least 4/5 mice survived for at least 60 days without trypanosome relapse. All other results are coded as resistant (R).

**Table 3. Results of drug sensitivity evaluation of *T b rhodesiense* clones in the mouse model.**

| Stab. No KETRI | Virulence Class | Pentamidine Mice cured/ total treated | | Melarsoprol Mice cured/ total treated | | diminazene aceturate Mice cured/ total treated | | Suramin Mice cured/ total treated |
|---|---|---|---|---|---|---|---|---|
| | | 5mg/kg | 20mg/kg | 1mg/kg | 20mg/kg | 2.5mg/kg | 20mg/kg | 2.5mg/kg |
| 2482 | Very-acute | 5/5 | 5/5 | 5/5 | 5/5 | 4/5 (1) | 5/5 | 2/2 (3)b |
| 3304 | | 5/5 | 5/5 | 5/5 | 5/5 | 5/5 | 5/5 | 5/5 |
| 3803 | | 4/4 (1)b | 5/5 | 5/5 | 4/4(1)b | 5/5 | 5/5 | 2/2(3)b |
| 2487 | Acute | 4/5 (1)a | 5/5 | 5/5 | 5/5 | 5/5 | 5/5 | 5/5 |
| 3801 | | 5/5 | 5/5 | 5/5 | 5/5 | 5/5 | 5/5 | 5/5 |
| 3798 | Sub- acute | 5/5 | 5/5 | 5/5 | 5/5 | 5/5 | 5/5 | 5/5 |
| 3926 | | 4/5(1)a | 5/5 | 5/5 | 5/5 | 5/5 | 5/5 | 3/3(2)b |
| 3380 | Chronic | 5/5 | 5/5 | 5/5 | 5/5 | 5/5 | 5/5 | 5/5 |
| 3928 | | 4/4 (1)b | 4/4 (1)b | 4/4 (1)b | 5/5 | 1/1(4)b | 5/5 | 5/5 |

Key: The mice were treated with single doses of various anti-trypanosomal drugs at 24 hours post infection and monitored for 60 days post treatment; a, number of mice which relapsed in each group during the 60 days of post-treatment monitoring; b, number of mice which died without parasitaemia relapse. All the isolates recorded at least 80% cure rates to all drug dose regimens and were therefore classified as sensitive.

give a concentration of 4mg/ml, whereas Melarsoprol (5 Ml vials of 180 mg) was first prepared by mixing (vortex) 1 Ml of the stock solution to 4 Ml of 50% propylene glycol to give a concentration of 7.2mg/ml (72mg/kg). This was further diluted to 40mg/kg by mixing(vortex) 5.6 ml of the 7.2mg/ml with 4.4 Mls of 50% propylene glycol to give a concentration of 4mg/ml (40mg/kg) The drug solutions for the 40mg/kg dose of each drug were then diluted serially using distilled water to give dosages for the 20, 10, 5, 2.5, 2, 1mg/kg.

## Statistical analysis

Analysis was carried out to test if there are significant differences between the four classes of virulence using PP, parasitaemia progression, PCV, body weight changes and survival as the response variables. The data obtained from the study were summarized using descriptive statistics. The general linear model in SAS was used to test significance at $p < 0.05$ level, between the means of the four virulence classes. Survival data analysis was carried out employing the Kaplan–Meier method on StatView (SAS Institute, Version 5.0.1) statistical package for determination of survival distribution function. Rank tests of homogeneity were used to determine the effect of virulence on host survival time [29].

## Results

### Survival time and classification

All control mice survived to the end of the experimental period of 60 days and their data (mean survival period) were therefore categorized as censored. The 12 *T. b. rhodesiense* clones exhibited variation in survival time and were classified into four classes of virulence based on these survival time data as shown (Table 1). A total of 3/12 clones were very acute, 3/12 were acute, 2/12 sub-acute and 4/12 chronic (Table 1).

The mean survival time (MST) of isolates categorized as very acute ranged from a mean ± SEM of 8.7±0.2 days (KETRI 2482) to 9.0 ± 0.4 for KETRI 3304 (Table 1, S1 Fig (i)). These survival time data were not statistically different from each other ($p > 0.05$). Mice infected with the acute clones had MST ranging from a mean±SEM of 18.1±0.7 (KETRI 2487) to 26 ±2.0 for *T.b rhodesiense* clone KETRI 3800 (S1 Fig (ii)); these data were significantly ($p < 0.0001$) different from each other (S1 Fig (ii)). In mice infected with *T. b. rhodesiense* clones characterized as sub-acute, the survival time ranged from a mean ± SEM of 28.2±2.0 for KETRI 3798 to mean ±SEM of 38.9 ± 1.6 for KETRI 3926 (S1 Fig (iii)) and were significantly ($p<0.0001$) different from each other. Mice infected with the *T. b. rhodesiense* clones characterized as chronic had MST values ranging from a mean±SEM of 45.6±1.6 for KETRI 3664 to a mean ± SEM of 55.5 ± 2.3 days for KETRI 3380 (S1 Fig (iv)) and were significantly ($p<0.001$) different from each other.

The survival time data for all the isolates in each virulence class were grouped together for the purpose of comparison among the groups. The overall MST was 8.7±0.2 for the very acute clones was, 21.6±1.0 for the acute clones 33.6±1.4 for the subacute clones and 50.0±1.8 for the chronic clones (Table 1, Fig 1). The Wilcoxon and Logrank tests had a p-value of 0.001, each showing that the MST of isolates in the different classes were significantly different.

### Pre-patent period and parasitaemia progression

The mean ±SEM pre-patent period in mice infected with very acute isolates ranged between 4 ±0.0 (KETRI 3803) to 5±0.0 (KETRI 2482 and 3304) and 4.7±0.09 when isolates are considered as a group (Table 1). The mean pre-patent period in mice infected with the acute class of the isolates ranged between 4±0.0 (KETRI 2487) and 6.0±0.0 (KETRI 3801) and 5.03±0.2 when they are considered as a class. With mice infected with sub-acute clones, the mean pre-patent

period ranged between 5.0±0.0 (KETRI 3926) and 5.3±0.2 (KETRI 3798) and 5.2±0.08 when considered as a class. The pre-patent period for the mice infected with the chronic isolates was in the range of 5.9±0.5 (KETRI 3380) to 6.7±0.2 (KETRI 3305) and 6.3±0.2 dpi when considered as a class (Table 1). When the pp data were grouped together for isolates in each virulence class, the mean ±SEM were 4.7±0.09 (very acute class), 5.03±0.2 (acute class), 5.2±0.08 (sub-acute class and 6.3±0.2 (chronic class). Despite the apparently increasing trend of these data, these differences were not statistically significant (p>0.05).

The parasitaemia patterns, results of the mean peak parasitaemia (±SE), and time (days) to peak parasitaemia (DPP) of the *T. b. rhodesiense* clones as shown in (Table 1, Fig 2). In mice infected with the three (3) very-acute clones, the mean parasitaemia of each clone was characterized by a single wave as compared to two waves for mice infected with the acute, subacute and chronic clones (S2 Fig (i); S2 Fig (ii), S2 Fig (iii) and S2 Fig (iv)).

*Trypanosoma b. rhodesiense* clone specific variation was observed in the mean peak parasitaemia and time (days) to peak parasitaemia (Table 1). In the very acute virulence class, clone KETRI 3803 attained the highest mean peak parasitamia of 9.3 x $10^8$ trypanosmes/mL of blood as compared to 1 x $10^6$ trypanosomes/mL of blood for clone KETRI 2482 (Table 1), a 100-fold difference; these differences were statistically significant (p < 0.001). The KETRI 3803 clone attained peak parasitaemia faster (6 days) as compared to 8 days for the KETRI 2482 (Table 1). The third clone in this class, KETRI3304 was intermediate with respect to both parameters. Out of the three clones classified as acute, KETRI 3800 and 3801 had significantly (p 0.001) lower mean peak parasitamia parasitaemia when compared to KETRI 2487 (S2 Fig (ii)). A comparable pattern was observed for the sub-acute clones, with KETRI 3926 showing a significantly (p<0.001) lower parasitaemia than clone KETRI 3798 (S2 Fig (iii)). Similarly, clone KETRI 3305 recorded significantly lower parasitaemia (S2 Fig (iv)) when compared to the rest of the *Tbr* clones classified as chronic.

When parasiataemia data for *T. b. rhodesiense* clones in each virulence class were grouped together and compared, parasitaemia was significantly (p<0.001) higher in mice infected with very- acute clones (Fig 2).

## Packed cell volume (PCV)

The pre-infection PCV data for all infected and control mice groups (Fig 3) were not statistically different (p > 0.05). The PCV of the infected mice declined significantly (p < 0.001)

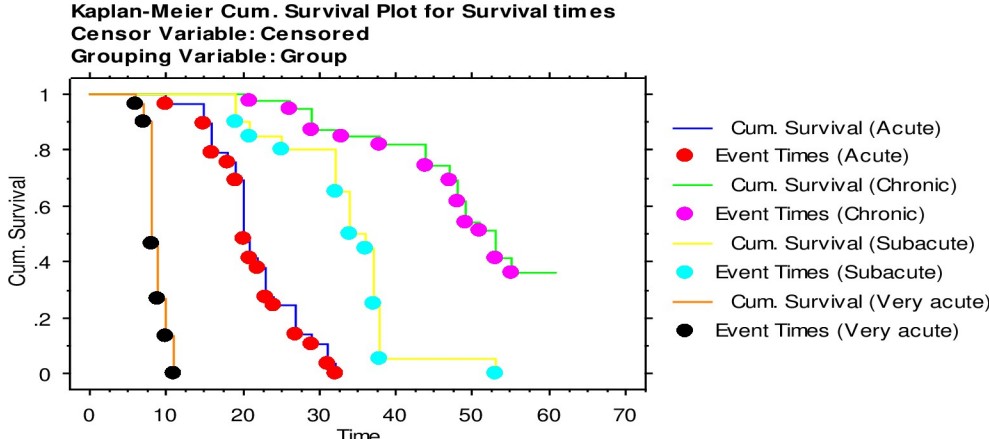

**Fig 1. The survival times for mice (n = 10) infected with twelve *T. b. rhodesiense* clones.** The clones were classified as (i) very-acute (0–15 dpi), (ii) acute (16–30 dpi), (iii) sub-acute (31–45 dpi), (iv) and chronic *Tbr* (46–60 dpi) all classes of virulence grouped together.

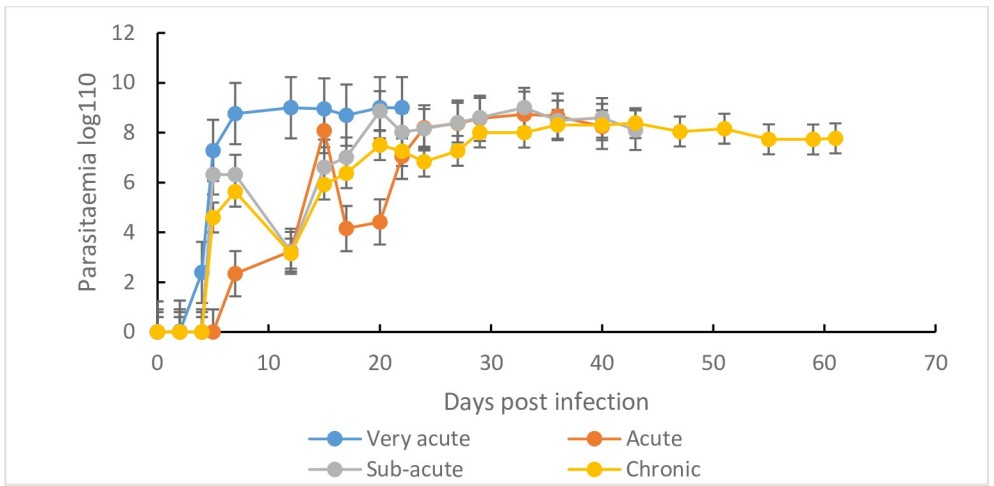

**Fig 2. Parasitaemia progression in mice (n = 10) infected with the four classes of *T. b. rhodesiense* clones.** (i) Very-acute, (ii) acute, (iii) sub-acute and (iv) chronic clones all classes of virulence grouped together.

following infection when compared with the PCV of the non-infected control mice which remained largely constant throughout the duration of the study (Fig 3). However, the onset and severity of the anemia, as shown by the decline in PCV, was most prominent for mice infected with the isolates classified as very acute (Fig 3). In these mice, the PCV declined significantly (p<0.001), from 49.7±0.8 at baseline (day 0) to 26.0±0.5 at 14 dpi equivalent to 47.7% decline. The lowest infection-related decline in PCV (Fig 3) was recorded in the mice infected with isolates classified as chronic clones, with the PCV declining from 49.6±0.9at baseline to 43.5±1.0at 14 dpi (12.3%).

The changes in PCV varied significantly between *T. b. rhodesiense* clones of the same virulence class (Fig 3). In mice infected with very-acute clones, KETRI 3304 infected mice recorded the highest decline from 48.25±1.6 at day 0 to 39± 2.4 at 7 dpi equivalent to 19.2% (S3 Fig (i)) whereas within the acute clones infected mice, KETRI 2487 infected mice recorded the highest decline from 47.7±1.8 at day 0 to 38.8±1.4 at 7 dpi equivalent to 18.7% (S3 Fig (ii)). In mice

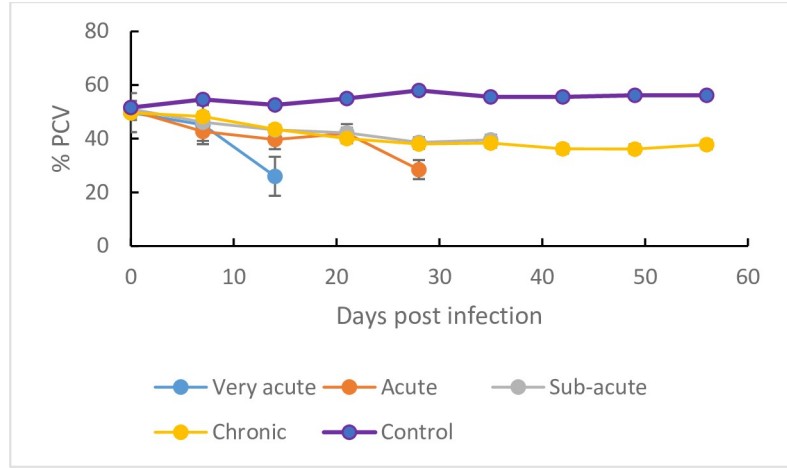

**Fig 3.** Mean ± SE PCV decline in mice (n = 10) infected with *T.b. rhodesiense* (i) very-acute isolates, (ii) acute isolates, (iii) sub-acute isolates and (iv) chronic isolates clones all classes of virulence grouped together.

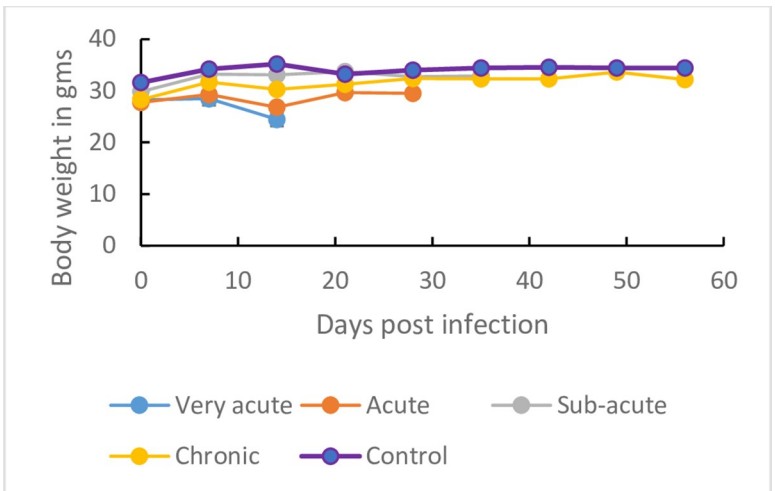

**Fig 4. Mean ± SE body weight changes in mice (n = 10) infected with *T.b. rhodesiense*.** (i) very-acute isolates, (ii) acute isolates, (iii) sub-acute isolates and (iv) chronic isolates all classes of virulence grouped together.

infected with the sub-acute clones, KETRI 3978 recorded the highest decline from 53.4±1.1 at day 0 to 47.9±0.6 at 7dpi equivalent to 10.3% (S3 Fig (iii)) and in mice infected with the chronic clones, KETRI 3305 infected mice registered the highest decline from 54±2 at day 0 to 47±1.4 at 7dpi equivalent to 13% (S3 Fig (iv)).

When the mean PCV data of all isolates in each virulence class were grouped together and compared, the decline in PCV was significantly ($p<0.001$) higher in mice infected with very-acute clones (Fig 3); the PCV of mice infected with the very acute clones declined from 49.7 ±0.8 to 26.0±0.5 at 14 DPI by an average of 47.7% as compared to an average decline of 12.3% for mice infected with the chronic *Tbr* clones.

## Body weight

The mean ±SE pre-infection body weight data for infected and control groups (Fig 4) were not statistically different ($p > 0.05$). Between 7 and 14 days post infection, all infected mice groups exhibited a decline in mean body weight (S4 Fig (i)–S4 Fig (iv)) while the body weight of un-infected control mice did not change (Fig 4). However, mice groups infected with clones classi-fied as acute, sub-acute and chronic exhibited recovery of their body weights starting 14 dpi. Mice group infected with isolates in the very acute virulence class did not survive beyond 14 dpi (Fig 4).

## Drug sensitivity results

The results of the drug-sensitivity testing for the reference sensitive (*Tbr* KETRI 3738) and drug-resistant (Tbr KETRI 2538) isolates are shown in (Table 2). The reference drug-resistant isolate was confirmed to be resistant to melarsoprol, Pentamidine and Diminazene aceturate at dose rates ranging from 1–20 mg/kg body weight (Table 2). However, infected mice were cured with all three drugs at a dose rate of 40 mg/kg body weight. With respect to suramin, the reference resistant isolate was sensitive to all doses equal to or greater than 5 mg/kg body and is therefore characterized as sensitive (Table 2). On the other hand, reference drug-sensitive isolate was confirmed to be sensitive to all doses of melarsoprol, ranging from 1–40 mg/kg bwt. It was also fully sensitive to diminazene aceturate at dose rates ranging from 2.5-40mg/kg bwt. The reference sensitive isolate was sensitive to pentamidne at all doses above 4 mg/kg bwt

(Table 2). It was also sensitive to all doses of suramin equal to or greater than 2.5 mg/kg (Table 2)

The results of drug sensitivity experiments for the test *Tbr* clones are summarized in (Table 3). All the isolates recorded at least 80% cure rates for all drug dose regimens evaluated in this study (Table 3) and were therefore classified as sensitive. However, a few cases of relapsewere observed in 1/5 (20%) mice infected with KETRI 2482 (very-acute group) and treated with diminazene aceturate at 2.5mg/kg, and KETRI 2487 (acute) and KETRI 3926 (sub-acute) treated with pentamidine at 5mg/kg. In mice infected with KETRI 3928 (Chronic), 4/5 treated with diminazene aceturate at 2.5mg/kg and 5/5 treated at 20 mg/kg died at 47 days post treatment and without detectable parasites (Table 3), This is presumably due to pneumonia resulting from cold environment caused by a leaking water bottle. No relapses were observed in mice groups that were treated with either melarsoprol (1 and 20 mg/kg) or suramin at 2.5 mg/kg (Table 2)

## Discussion

In this study, we characterized the virulence and anti-trypanosomal drug sensitivity patterns of 12 *T. b. rhodesiense* cloned stabilates. We used *T. b rhodesiense* clones because they represent a homogeneous population of genetically identical trypanosomes thus making subsequent studies based on these clones more reproducible and reliable as previously reported [30]. The results demonstrated the existence of variation in virulence of *T. b. rhodesiense* cloned stabilates (Table 1) which is interesting because all study isolates were originally recovered from western Kenya and eastern Uganda, regions that are considered to belong to the same Busoga focus of HAT. Our results are in agreement with a study on a number of isolates from eastern Uganda in mice which showed that distinct acute and chronic strains of *T. b. rhodesiense* circulated in the focus [31]. The results are also also in agreement with a previou report of variations in virulence of *T. b. gambiense* isolates [32].

We used mean survival time (MST) of mice post-infection as the main indicator of virulence as previously reported [26, 27, 33]; and observed that the *Tbr* isolates were well distributed among the four virulence classes. Infective isolates that allowed mice to have long survival times, hence chronic infections, may indicate presence of enriched population of stumpy forms which aids in prolonging host survival and enhancing the probability of parasite transmission [34]. The mean survival times for the very acute clones was 8.7 days suggesting the hosts were overwhelmed by the first parasitaemia peak before the proliferating slender forms differentiated into short stumpy forms [35]. The majority (11/12) of the *Tbr* clones used in this study had undergone a minimal number of (1–8) passages since isolation (Table 2). The only isolate that had undergone a significant (64) number of passages (Table 2) did exhibit notably different virulence characteristics from the other clones in the class, suggesting therefore that the observed differences in isolate virulence are an intrinsic attribute as previously reported [36].

Parasitaemia progression differed significantly among *Tbr* isolates assigned to different virulence classes on the basis of survival time. This finding is in agreement with previous studies in which virulence of different species of trypanosomes was characterised using parasitaemia, intensity of anaemia (PCV) and weight loss experienced by the host during the infection period [27]. In our study, parasitaemia of isolates in the very-acute virulence class was represented by a single wave whereas the acute, sub-acute and chronic virulence classes were represented by two waves. (Fig 2). This is in agreement with studies by [37] who observed that acute infections result from uncontrolled proliferation of the slender trypanosome forms without differentiation into short stumpy forms and hence kills the host before tsetse transmission

takes place [37]. In contrast, chronic infection is characterized by appearance of progressive waves of parasitaemia, with each distinct wave being composed of trypanosomes with antigenically distinct coats, and with parasites easily differentiating into the transmissible short stumpy forms. This perhaps explains why highly virulent trypanosomes are not easily transmissible as was observed by [22] that tsetse flies infected with chronic *T. b. brucei* recorded highest mature infection as opposed to those infected with highly virulent trypanosomes. Our results are important as they reveal that the majority of *T. b. rhodesiense* infections are in the bracket of (acute, sub-acute and chronic) classes of virulence and can easily be transmittable.

In the present study, all infected mice recorded a decline in PCV signifying the development of *T. b. rhodesiense* induced anemia. Our observation was in agreement with previous studies which reported anaemia as a key feature both in humans [38] and in the monkey model [6]. As with parasitaemia and survival time parameters, the development of anemia was significantly pronounced in mice infected with very-acute clones. This finding is consistent with observations by [39] who reported that acute infection of mice with *Trypanosoma cruzi* was characterized by an exponential growth of parasites and high mortality accompanied by anemia. A similar observation was made by [27] in mice infected with *Trypanosoma evansi*. In contrast, anaemia in mice infected with clones in the other various classes of virulence (acute, sub-acute and chronic) stabilized or recovered characteristic of the chronic phase anaemia [40]. The severity of anemia is determined by parasite virulence, time lag from infection to therapeutic intervention and individual host differences [41].

Our results showed a decline in body weight in the early days of infection (7–14 dpi)

This decline was however not significant. Our observation is important as it confirms a previous observation [22] that body weight alone cannot conclusively serve as a virulence biomarker. Previous authors [42] attributed decline in body weight to reduced food intake. In our study, we did not measure the food intake. The failure by infected mice to register a significant decline in body weight calls for further investigation on the causes of body weight changes in animals infected with trypanosomes especially after previous studies have recorded an increase in body weight in *T.evansi* [27] and in *T. b. brucei* or *T. congolense* [22] infected mice.

Our results on drug sensitivity tests showed that all the study isolates were sensitive to melarsoprol, pentamidine, diminazene aceturate and suramin. The sensitivity of these isolates to suramin and melarsoprol is significant since currently they are the only drugs of choice recommended by WHO (2018) to treat early and late stages of *Tbr* HAT respectively. On the other hand the sensitivity of the *Tbr* isolates to diminazene aceturate, is an indicator of the utility of these drug when administered to livestock reservoirs of *Tbr* isolates as practiced in disease HAT control programmes in endemic countries [43] Interestingly, however, the single cases of relapses encountered in mice infected with KETRI 2482 (very- acute virulence class), KETRI 2487 (acute virulence class) and 3926 (sub-acute virulence class) were all against the two diamidines (pentamidine or dimainazene) but not against suramin or melarsoprol (Table 3) which is consistent with clinical practice of not using these specific diamidines to treat *Tbr* HAT (WHO, 2018). Overall, the fact that the test isolates were all sensitive (at least 80% cure rates) to the drugs suggests there was no relationship between isolates' virulence and their sensitivity to anti-trypanosomal drugs which is in agreement with previous studies [44, 45]. In a study by Sokolova et al [46], these authors observed that resistance to nifurtimox did not compromise parasite virulence. This is despite previous studies reporting that drug resistant trypanosome have reduced virulence [47, 48].

The KALRO-BioRI reference isolate considered to be drug resistant was confirmed in this study to be resistant to melarsoprol, pentamidine and diminazene aceturate (Table 2). In general drug resistance is attributed to reduced drug uptake due the mutation or absence of a drug uptake gene [49] as well as by enhanced drug export, mediated by a multidrug resistance-

associated protein [50]. The uptake of the three drugs, melarsoprol, pentamidine and diminazene is mediated by the P2 transporter [12, 51, 52] which explains why resistance to all three drugs is linked. In contrast, uptake of suramin by trypanosomes is not mediated by the P2 transporter, hence the reason why the trypanosome, KETRI 2538, retains sensitivity to suramin

In summary, this study has found that there is variation in virulence of cloned stabilates made from field isolates recovered from western Kenya/eastern Uganda HAT focus. Virulence is attributed to the production by the blood stream forms of membranous nanotubes that originate from the flagellar membrane and disassociate into free extracellular vehicles (EVs). This (EVs) contain several flagellar proteins that contribute to virulence [53]. Since our study was based on laboratory cloned *T. b. rhodesiense* stabilates, however, future studies should utilize the parent primary isolates. Our results are important as they have demonstrated that virulence is not a hindrance in the control of trypanosomiasis by chemotherapy.

## Supporting information

**S1 Fig.** (i): Mean survival times in mice infected with the very-acute clones of *Trypanosoma brucei rhodesiense*. (ii): Mean survival times in mice infected with the acute clones of *Trypanosoma brucei rhodesiense*. (iii): Mean survival times in mice infected with the sub-acute clones of *Trypanosoma brucei rhodesiense*. (iv): Mean survival times in mice infected with the chronic clones of *Trypanosoma brucei rhodesiense*.
(DOCX)

**S2 Fig.** (i): Mean parasitaemia progression in mice infected with the very-acute clones of *Trypanosoma brucei rhodesiense*. (ii): Mean parasitaemia progression in mice infected with the acute clones of *Trypanosoma brucei rhodesiense*. (iii): Mean parasitaemia progression in mice infected with the sub-acute clones of *Trypanosoma brucei rhodesiense*. (iv): Mean parasitaemia progression in mice infected with the chronic clones of *Trypanosoma brucei rhodesiense*
(DOCX)

**S3 Fig.** (i): Mean PCV decline in mice infected with the very acute clones of *Trypanosoma brucei rhodesiense*. (ii): Mean PCV decline in mice infected with the acute clones of *Trypanosoma brucei rhodesiense*. (iii): Mean PCV decline in mice infected with the sub-acute clones of *Trypanosoma brucei rhodesiense*. (iv): Mean PCV decline in mice infected with the chronic clones of *Trypanosoma brucei rhodesiense*.
(DOCX)

**S4 Fig.** (i): Mean body weight changes in mice infected with the very acute clones of *Trypanosoma brucei rhodesiense*. (ii): Mean body weight changes in mice infected with the acute clones of *Trypanosoma brucei rhodesiense*. (iii): Mean body weight changes in mice infected with the sub-acute clones of *Trypanosoma brucei rhodesiense*. (iv): Mean body weight changes in mice infected with the very chronic clones of *Trypanosoma brucei rhodesiense*.
(DOCX)

## Acknowledgments

We acknowledge the Director, KALRO for permission to publish this study. Our other acknowledgment goes to Dr. Johnson Ouma, former Center Director (Trypanosomiasis Research Center) BioRI for supervision and facilitation, technical staff of KALRO- BioRI and in particular John Ndichu, Jane Hanya for taking care of the infected mice. Gilbert Ouma and Mr. Mageto for the preparation of drugs.

## Author Contributions

**Conceptualization:** Kariuki Ndung'u, Grace Adira Murilla.

**Data curation:** Kariuki Ndung'u, Purity Kaari Gitonga, Daniel Kahiga Thungu.

**Formal analysis:** Geoffrey Njuguna Ngae.

**Funding acquisition:** Grace Adira Murilla.

**Investigation:** Kariuki Ndung'u, Grace Adira Murilla, Joanna Eseri Auma, Purity Kaari Gitonga, Daniel Kahiga Thungu.

**Methodology:** Richard Kiptum Kurgat.

**Supervision:** Kariuki Ndung'u, Raymond Ellie Mdachi.

**Writing – original draft:** Kariuki Ndung'u, Joanna Eseri Auma.

**Writing – review & editing:** Grace Adira Murilla, John Kibuthu Thuita, Judith Kusimba Chemuliti, Raymond Ellie Mdachi.

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
