## [Decision Letter · Decision Letter 0]

26 Feb 2020

PONE-D-20-02583

Differential virulence of Trypanosoma brucei rhodesiense isolates does not influence the outcome of treatment with anti-trypanosomal drugs in the mouse model

PLOS ONE

Dear Mr. Ndungu,

Thank you for submitting your manuscript to PLOS ONE. After careful consideration, we feel that it has merit but does not fully meet PLOS ONE’s publication criteria as it currently stands. Therefore, we invite you to submit a revised version of the manuscript that addresses the points raised during the review process.

Please attend to all the comments from the reviewers and in addition, it would add more value if you could attend to the following as well:

I) Why was the control group (not infected mice) not included in Table 1. Although this group is not mentioned in Materials and Methods, it is mentioned in results in lines 200 and 213. This would add more credence to the results.

ii) In line 188, please mention the groups that the mice infected with very acute clone were being compared to.

iii) Line 52, add a full stop after "infection" and "an" before "infection" 

We would appreciate receiving your revised manuscript by Apr 11 2020 11:59PM. To enhance the reproducibility of your results, we recommend that if applicable you deposit your laboratory protocols in protocols.io, where a protocol can be assigned its own identifier (DOI) such that it can be cited independently in the future. For instructions see: http://journals.plos.org/plosone/s/submission-guidelines#loc-laboratory-protocols

We look forward to receiving your revised manuscript.

Kind regards,

Martin Chtolongo Simuunza, PhD

Academic Editor

PLOS ONE

Journal Requirements:

2. We noticed you have some minor occurrence(s) of overlapping text with the following previous publication(s), which needs to be addressed:

https://doi.org/10.1371/journal.pntd.0001857

https://doi.org/10.1017/S0031182017002359

https://www.afro.who.int/health-topics/trypanosomiasis-african

In your revision ensure you cite all your sources (including your own works), and quote or rephrase any duplicated text outside the Methods section. Further consideration is dependent on these concerns being addressed.

4. Please upload a new copy of Figure 1 as the detail is not clear. Please follow the link for more information: http://blogs.PLOS.org/everyone/2011/05/10/how-to-check-your-manuscript-image-quality-in-editorial-manager/

Reviewers' comments:

Reviewer's Responses to Questions

**Comments to the Author**

1. Is the manuscript technically sound, and do the data support the conclusions?

Reviewer #1: Yes

Reviewer #2: Yes

2. Has the statistical analysis been performed appropriately and rigorously? 

Reviewer #1: No

Reviewer #2: Yes

3. Have the authors made all data underlying the findings in their manuscript fully available?

Reviewer #1: No

Reviewer #2: Yes

4. Is the manuscript presented in an intelligible fashion and written in standard English?

Reviewer #1: Yes

Reviewer #2: Yes

5. Review Comments to the Author

Reviewer #1: This is an interesting comparative paper in which 12 clones are compared. These clones all originate from stabilates that were originally collected in the same HAT area, which adds particular value to the paper. However, there are some issues that dampen the value of the paper as it lacks scientific data that could be very valuable for others. There is also one scientific concept that according to this reviewer should be taken into account for the discussion section.

Main problem:

Figures 1 to 4 have data that combines observations from different clones grouped together. Because there is no specific information on the 12 clones, all scientific data is lost, and no external reader can see what exactly is going on. All figures should be split into the 4 groups described, and for each group, all individual clone data should be provided. The figure legend should include how many mice were used to obtain data for each clone, and how many repeats were done. Without this information, the paper is just a 'cartoon'. With this data, this reviewer is convinced the paper could have a big impact and could open the door even for collaborative work, as it seems the authors have a very valuable unique collection in their hands. But, for that the scientific data has to be there so it is clear for every clone what is exactly happening.

Interpretation issue:

The authors started with 12 stabilates and selected from every stabilate a single clone. In the discussion, they 'walk' backwards and suggest that the individual virulence of each clone, could reflect different levels of virulence of human infections. In order to do that, a crucial experiment is missing. In order to make such conclusion one should take different clones from one and the same , and show that within a stabliate, all clones have the same level of virulence. This is actually very unlikely because it is known that every field stabilate contains a whole collection of different parasites expressing different VSGs and possibly even using different expression sites. Taking one clone per stabilate, and just assuming that this clone represent the behavior of the entire population has no scientific basis at all. Hence, any conclusion based on such assumption is juts scientifically wrong. Please remove that speculation section from the discussion because it undermines the intellectual value of the paper.

Reviewer #2: This is a nicely written manuscript describing a study of virulence and drug resistance in 12 isolates of the human-infective kinetoplastid parasite Trypanosoma brucei rhodesiense, which causes the acute East African form of sleeping sickness. In brief, the authors assessed the virulence of each isolate by infecting groups of 10 male Swiss White mice and measuring survival times, parasitemia progression and changes in body weight and packed cell volume. They also used mice to determine the sensitivity of each isolate to several commonly used HAT drugs, including Melarsoprol, Diminazene acetate, Pentamidine and Suramin, administered at doses ranging from 1 - 40 mg/kg body weight. Their main result is that despite the existence of substantial variation in virulence amongst isolates, all 12 isolates were highly sensitive to each of these drugs, even when administered at the low doses.

As I am not an epidemiologist, I am not in a position to judge the clinical significance of this work. However, I have a few concerns, possibly minor, that I would encourage the authors to address.

1.) It would be helpful to non-specialist readers if the authors included a few sentences remarking on the clinical relevance (or lack thereof) of the mouse model for virulence and drug resistance of T. brucei in humans. For example, is there any reason to expect that isolates that are exceptionally virulent in mice will be also be exceptionally virulent in humans and vice versa? The authors do demonstrate that Tbr isolates previously characterized as sensitive or resistant remain so using the mouse assays described in this paper, but similar controls are not provided for virulence.

2.) In lines 277-278, the authors remark that there is no correlation between virulence and number of passages since isolation and they cite Table 2 as evidence, but I fail to see how the information in Table 2 makes this point. Furthermore, it would appear that the very acute isolates are mostly older than the less acute isolates. Is there evidence of a reduction in the virulence of Tbr in recent years, perhaps because of enhanced genetic drift as the number of Tbr infections decreases?

3.) In lines 236-238, the authors remark that several of the mice treated with diminazene acetate died "due to causes not related to trypanosome infection." Please provide more details, e.g., what was the cause of death? In light of the small number of mice (5) used to assess resistance, it is somewhat concerning that so many of them (14) died from unrelated causes.

4) This study suggests that virulence is not a strong predictor of drug resistance for Tbr sampled over this time scale from these particular locations. However, the significance of this result is somewhat weakened by the fact that none of the tested isolates exhibited resistance. Are there data from other locations that suggest that such a correlation sometimes exists?

5.) In Table 2, Sensitive Isolate KETRI 2537 is shown as resistant to MelB at the lowest drug does (1 mg/kg) despite the fact that all 5 mice are shown as being cured. Is this a data entry error? If not, pleas explain the result.

There are several places in the text where the wording could be improved. These are listed below with the corresponding line numbers:

line 45: two subspecies

line 47: tsetse flies

line 53: two stages, namely the early (hemolymphatic) and the late (meningo-encephalitic) stage.

line 59: as well as anemia

line 72: In their study, Pyana and colleagues [13]

line 86: 6- to 8-week old

line 114: What do you mean by "ip"?

line 130: The classification of trypanosome virulence was based

line 141: determined for Melarsoprol

lines 149 and 151: 10 ml

line 160: if there are

line 170: exhibited variation

line 185: were not statistically significant

line 258: [27] does not appear to be the correct reference for this statement about Tbr genetic variation

line 259: the existence of variation in virulence among

line 261: Uganda

line 310: Our results showed a decline in body weight in the early days of infection (7 - 14 dpi) followed by a recovery except in mice infected with very acute strains.

line 312: confirms a previous observation

line 318: I'm not sure what you mean by "with days post infection"?

line 341: there is variation in virulence

line 348: Other studies are needed to confirm

6. PLOS authors have the option to publish the peer review history of their article (what does this mean?). If published, this will include your full peer review and any attached files.

Reviewer #1: No

Reviewer #2: Yes: Jay Taylor

---

## [Author Response · Author response to Decision Letter 0]

29 May 2020

#

 Reviewers comments Response

 Why was the control group (not infected mice) not included in Table 1. Although this group is not mentioned in Materials and Methods, it is mentioned in results in lines 200 and 213. This would add more credence to the results. We concur with the reviewer’s suggestion and have therefore added the following sentence to the materials and methods section “A control group of 10 non-infected mice was included in the study and monitored for changes in PCV, body weight and survival times” ”margin lines 127-128. We have also included the control group in table 1 of the marked manuscript margin line 277-278”

 In line 188, please mention the groups that the mice infected with very acute clone were being compared to. We concur with the reviewer. In the revised manuscript, this statement has been re-phrased to read as follows: “When parasiataemia data for T. b. rhodesiense clones in each virulence class were grouped together and compared, parasitaemia was significantly (p<0.001) higher in mice infected with very- acute clones, ”margin line 253-256 of the marked manuscript.

 We noticed you have some minor occurrence(s) of overlapping text with the following previous publication(s), which needs to be addressed:

https://doi.org/10.1371/journal.pntd.0001857
https://doi.org/10.1017/S0031182017002359
https://www.afro.who.int/health-topics/trypanosomiasis-african

We have made changes to the manuscript to address the issue of overlapping texts. Where the text in question needs to be maintained, we have provided appropriate citations as follows: 

1) https://doi.org/10.1371/journal.pntd.0001857. 

We have modified our text to read as follows:

According to Maclean et al (2008), the pathology encountered in the acute HAT infections is characterized by elevated Tumor necrosis factor alpha (TNF-α) while that encountered in the chronic HAT infections is characterized by elevated transforming growth factor (TGF-β) (Line 66-68)

2) https://doi.org/10.1017/S0031182017002359

In the revised manuscript, the affected section now reads as follows “Each mouse’s survival time was determined on the basis of a ≥ 25% decline in PCV and consistently high parasitaemia levels of 1x109/ml for at least two consecutive days as previously described by Kamidi et al,2018 [26]” (margin lines 141-146 of the marked manuscript).The work by Kamidi et al, 2018 was done in the same laboratory in which the present study was done, hence similar methods were used to determine survival time. 

3. https://www.afro.who.int/health-topics/trypanosomiasis-african

We have improved our text to read ‘’ The sensitivity of these isolates to suramin and melarsoprol is significant since currently they are the only drugs of choice recommended by WHO (2018) to treat early and late stages of Tbr HAT respectively’’. Lines 4324-436 of the marked manuscript

 Please amend your list of authors on the manuscript to ensure that each author is linked to an affiliation. Authors’ affiliations should reflect the institution where the work was done (if authors moved subsequently, you can also list the new affiliation stating “current affiliation:….” as necessary) We acknowledge this observation by the reviewer and has hence forth corrected this in the revised manuscript. “Margin lines 8,13 and 15 of the marked manuscript”

 Please upload a new copy of Figure 1 as the detail is not clear. Please follow the link for more information: http://blogs.PLOS.org/everyone/2011/05/10/how-to-check-your-manuscript-image-quality-in-editorial-manager/

A new copy of this figure has been uploaded

 Please include captions for your Supporting Information files at the end of your manuscript, and update any in-text citations to match accordingly. Please see our Supporting Information guidelines for more information: http://journals.plos.org/plosone/s/supporting-information.

 We concur with the reviewer. Supporting information captions are now included at the end of the manuscript as suggested.

 Reviewer # 1 

1 Figures 1 to 4 have data that combines observations from different clones grouped together. Because there is no specific information on the 12 clones, all scientific data is lost, and no external reader can see what exactly is going on. All figures should be split into the 4 groups described, and for each group, all individual clone data should be provided. The figure legend should include how many mice were used to obtain data for each clone, and how many repeats were done. Without this information, the paper is just a 'cartoon'. With this data, this reviewer is convinced the paper could have a big impact and could open the door even for collaborative work, as it seems the authors have a very valuable unique collection in their hands. But, for that the scientific data has to be there so it is clear for every clone what is exactly happening. We appreciate this observation by the reviewer. For each of the figures 1-4, we have now provided data for each of the clones as supporting information as follows: survival (Fig 1(i)-(iv) in S1, Parasitaemia Fig 2 (i)-(iv) in S2, PCV Fig 3 (i)-(iv) in S3 and body weight Fig 4(i)-(iv) in S4 and updated the relevant sections of the text as follows: survival data (margin lines 185-203), parasitaemia (margin lines 234-256), PCV (margin lines 293-306) and body weight (margin lines 311-317) of the revised manuscript.

The number of mice used in each virulence study is now provided in the legend as was suggested by the reviewer (survival, line 203, parasitaemia, line 257, PCV, line 307 and body weight, line 318) 

2 The authors started with 12 stabilates and selected from every stabilate a single clone. In the discussion, they 'walk' backwards and suggest that the individual virulence of each clone, could reflect different levels of virulence of human infections. In order to do that, a crucial experiment is missing. In order to make such conclusion one should take different clones from one and the same, and show that within a stabilate, all clones have the same level of virulence. This is actually very unlikely because it is known that every field stabilate contains a whole collection of different parasites expressing different VSGs and possibly even using different expression sites. Taking one clone per stabilate, and just assuming that this clone represent the behavior of the entire population has no scientific basis at all. Hence, any conclusion based on such assumption is juts scientifically wrong. Please remove that speculation section from the discussion because it undermines the intellectual value of the paper.

 We thank the reviewer for this comments. We have addressed the reviewer’s comments in two ways:

1) Expanding on our explanation why we used cloned stabilates instead of field stabilates. In the revised manuscript, the sentence reads “We used T. b rhodesiense clones because they represent a homogeneous population of genetically identical trypanosomes thus making subsequent studies based on these clones more reproducible and reliable as previously reported (lines 359-361)

2) We have deleted reference to clinical profiles of HAT as suggested by the reviewer (margin lines 372-373). 

3) We have also included the statement ’However, we recommend a similar study be carried out using the parent primary isolates (margin lines (463-465) of the revised manuscript.

 Reviewer # 2 

1 It would be helpful to non-specialist readers if the authors included a few sentences remarking on the clinical relevance (or lack thereof) of the mouse model for virulence and drug resistance of T. brucei in humans. For example, is there any reason to expect that isolates that are exceptionally virulent in mice will be also be exceptionally virulent in humans and vice versa? The authors do demonstrate that Tbr isolates previously characterized as sensitive or resistant remain so using the mouse assays described in this paper, but similar controls are not provided for virulence.

 We concur with the reviewer’s suggestions and have therefore inserted in the statement “Studies on disease pathogenesis, parasite virulence, drug sensitivity and identification of new potential drug targets and staging biomarkers are commonly carried out in the mouse model based on its cost effectiveness, genetic similarity to humans estimated to be 97.5%, and ethical limitations of carrying out such studies in higher animal models or humans [14,15,16] .” margin lines 80 -84 of the marked manuscript

Direct comparison of parasite virulence data obtained using inbred mice (minimal host variation) and data obtained from clinical cases of HAT is difficult due to host variations in humans. Therefore, since the use of inbred mice is designed to minimize/eliminate the effect of host variations on disease outcome, it is common to infer that parasites found to be virulent in mouse model would also be virulent in humans. 

No controls were provided for virulence because the idea for the current study was to compare parasite virulence with drug sensitivity patterns. 

2 In lines 277-278, the authors remark that there is no correlation between virulence and number of passages since isolation and they cite Table 2 as evidence, but I fail to see how the information in Table 2 makes this point. Furthermore, it would appear that the very acute isolates are mostly older than the less acute isolates. Is there evidence of a reduction in the virulence of Tbr in recent years, perhaps because of enhanced genetic drift as the number of Tbr infections decreases? We appreciate the reviewers comment. We would, however, like to clarify that it was never our intention to state that there is no correlation between virulence and number of passages. Rather, the intended meaning was that since the study isolates had undergone only a minimal number of passages, the observed differences in virulence are likely caused by intrinsic attributes of the trypanosome. In the revised manuscript, the relevant sections reads as follows: “The majority (11/12) of the Tbr clones used in this study had undergone a minimal number of passages (1-8) passages since isolation (Table 2). The only isolate that had undergone a significant (64) number of passages (Table 2) did not exhibit notably different virulence characteristics from the other clones in the class, suggesting therefore that the observed differences in isolate virulence is an intrinsic attribute as previously reported [36] (lines 378-383) of the revised manuscript. Table 1 has been modified to show the number of passages of each trypanosome isolate (margin line 277-278). There is no data comparing virulence between older and recent Tbr isolates and there will be need to undertake such a study in future.

3 In lines 236-238, the authors remark that several of the mice treated with diminazene acetate died "due to causes not related to trypanosome infection." Please provide more details, e.g., what was the cause of death? In light of the small number of mice (5) used to assess resistance, it is somewhat concerning that so many of them (14) died from unrelated causes. We are in agreement with the reviewer and have rephrased the sentence to read as follows:…. died at 47 days post treatment and without parasitologically detectable trypanosome (Table 3), This is presumably due to pneumonia resulting from cold environment caused by a leaking water bottle. “margin lines 339-342 of the marked manuscript”

4 This study suggests that virulence is not a strong predictor of drug resistance for Tbr sampled over this time scale from these particular locations. However, the significance of this result is somewhat weakened by the fact that none of the tested isolates exhibited resistance. Are there data from other locations that suggest that such a correlation sometimes exists? We appreciate this concern raised by the reviewer. This has now been improved in the revised manuscript and reads as follows” Overall, the fact that the test isolates were all sensitive (at least 80% cure rates) to the drugs suggests there was no relationship between isolates’ virulence and their sensitivity to anti-trypanosomal drugs which is in agreement with previous studies [46,47]. In a study by Sokolova et al, [48]these authors observed that resistance to nifurtimox did not compromise parasite virulence. This is despite previous studies reporting that drug resistant trypanosome have reduced virulence [49,50]. (margin lines 444-449).

5 In Table 2, Sensitive Isolate KETRI 2537 is shown as resistant to MelB at the lowest drug does (1 mg/kg) despite the fact that all 5 mice are shown as being cured. Is this a data entry error? If not, please explain the result. We welcome this observation by the reviewer and has as a results corrected our data to read 0/5 implying all the 5 mice were resistant to MelB 1mg/kg (margin line 345-346) of the marked manuscript.

line 45: two subspecies This is now corrected as suggested by the reviewer “margin line 47 of the marked manuscript”

line 47 tsetse flies This now corrected as suggested by the reviewer “margin lines 49 of the marked manuscript”

line 53 two stages, namely the early (hemolymphatic) and the late (meningo-encephalitic) stage. This now corrected as suggested by the reviewer “margin line 55 of the marked manuscript”

line 59 as well as anemia Corrected as suggested by the reviewer “margin line 61 of the marked manuscript”

line 72 In their study, Pyana and colleagues [13] Corrected as suggested by the reviewer “margin lines 74-75 of the marked manuscript”

line 86 6- to 8-week old Corrected as suggested “margin line 93 of the marked manuscript”

line 114 What do you mean by "ip"? This is an abbreviation of intraperitoneal as described on margin line 112 of the marked manuscript

line 130 The classification of trypanosome virulence was based Corrected as suggested “margin line 138 of the marked manuscript”

line 141 determined for Melarsoprol

 Corrected as suggested “margin line 151 of the marked manuscript”

lines 149 and 151 10 ml Corrected as suggested “margin lines 161, 162 and 165 of the marked manuscript”

line 160 if there are Deleted the word exist and rephrased the sentence to read ….. if there are significant differences….. “margin line 170 of the marked manuscript”

line 170 exhibited variation Corrected as suggested “margin line 182 of the marked manuscript”

line 185 were not statistically significant Corrected as suggested by the reviewer “margin line 232 of the marked manuscript”

line 258 [27] does not appear to be the correct reference for this statement about Tbr genetic variation We wish to clarify that this reference [27] is in the material and method section on determination of survival and has no reference to genetic variation (margin lines 145 -146 of the marked manuscript).Statement on genetic variation is in the discussion section and is supported by reference [30] margin line 360-361.

line 259 the existence of variation in virulence among Corrected as suggested “margin line 362 of the marked manuscript”

line 261 Uganda Corrected as suggested “margin line 364 of the marked manuscript”

line 310 Our results showed a decline in body weight in the early days of infection (7 - 14 dpi) followed by a recovery except in mice infected with very acute strains. Corrected as suggested “margin lines 422of the marked manuscript”

line 312 confirms a previous observation Corrected as suggested “margin line 426 of marked manuscript”

line 318 I'm not sure what you mean by "with days post infection"? We concur with the reviewer that this statement is ambiguous and is now deleted “margin line 432 of the marked manuscript”

line 341 there is variation in virulence Corrected as suggested “margin line 459 of the marked manuscript”

line 348 Other studies are needed to confirm

 This has been rephrased to read as follows “Since our study was based on laboratory cloned T. b. rhodesiense stabilates, however, future studies should utilize the parent primary isolates”.

 “ margin line 463-465 of the marked manuscript”

---

## [Decision Letter · Decision Letter 1]

12 Jun 2020

PONE-D-20-02583R1

Differential virulence of Trypanosoma brucei rhodesiense isolates does not influence the outcome of treatment with anti-trypanosomal drugs in the mouse model

PLOS ONE

Dear Dr. Ndungu,

Thank you for submitting your manuscript to PLOS ONE. After careful consideration, we feel that it has merit but does not fully meet PLOS ONE’s publication criteria as it currently stands. Therefore, we invite you to submit a revised version of the manuscript that addresses the points raised during the review process.

Please attend to the corrections and comments raised by Reviewer #2

We look forward to receiving your revised manuscript.

Kind regards,

Martin Chtolongo Simuunza, PhD

Academic Editor

PLOS ONE

Reviewers' comments:

Reviewer's Responses to Questions

**Comments to the Author**

1. If the authors have adequately addressed your comments raised in a previous round of review and you feel that this manuscript is now acceptable for publication, you may indicate that here to bypass the “Comments to the Author” section, enter your conflict of interest statement in the “Confidential to Editor” section, and submit your "Accept" recommendation.

Reviewer #1: All comments have been addressed

Reviewer #2: (No Response)

2. Is the manuscript technically sound, and do the data support the conclusions?

Reviewer #1: Yes

Reviewer #2: Yes

3. Has the statistical analysis been performed appropriately and rigorously? 

Reviewer #1: N/A

Reviewer #2: Yes

4. Have the authors made all data underlying the findings in their manuscript fully available?

Reviewer #1: Yes

Reviewer #2: Yes

5. Is the manuscript presented in an intelligible fashion and written in standard English?

Reviewer #1: Yes

Reviewer #2: No

6. Review Comments to the Author

Reviewer #1: Thank you for addressing the concerns flagged by this reviewer after assessment of the first version of the manuscript.

Reviewer #2: The authors have successfully addressed all of the major concerns raised in my original review and the

resulting study makes an interesting contribution to our understanding virulence and drug resistance in

African trypanosomiasis. However, there are still some minor grammatical errors and one questionable

claim in the revised manuscript, as listed below. Note that the line numbers given here refer to the

unmarked revised text in the file sent for review, not the marked revised text that appears in the second

half of the file.

line 91: The authors remark that the mouse has a "genetic similarity to humans estimated to be 97.5%",

but I don't know any way of defining genetic similarity that would give such a high number. According

to the website https://www.genome.gov/10001345/importance-of-mouse-genome, the average similarity

between the shared protein-coding genes is around 85%. Please delete or modify this claim.

line 45: A cure rate of at least 80% was achieved for all test isolates

line 48: evidence of variation in ... confirms that this variation is

line 73: in different foci differ in both their

line 87: this variation

line 143: once a week using a

line 146: The classification of trypanosome virulence

line 165: were prepared

line 179: body weight changes

line 181: The general linear model in SAS was ... between the means of the four virulence classes.

line 190: exhibited variation in

line 196: Mice infected with the acute clones had MST ranging from 18.1 +- 0.7 (KETRI 2487) to 26 +- 2.0 for

... (You don't need to keep repeating the phrase mean +- SEM for each set of numbers.)

line 207: The overall MST was 8.7 +- for the very acute clones, 21.6 +- 1.0 for the acute clones, ...

line 227: are shown in Table and Figure 2(v).

line 231: clone specific variation was observed

line 257: The pre-infection PCV data for all infected

line 258: The PCV of the infected mice declined significantly following infection when compared

line 308: at least 80% cure rates for all drug dose regimens

line 309: a few cases of relapse

line 314: without detectable parasites.

line 343: a previous report of variation in virulence

line 356: differences in isolate virulence are an intrinsic attribute

line 357: Parasitemia progression differed significantly among Tbr isolates assigned to different virulence

classes on the basis of survival time.

line 363: acute infections result from

line 371: they reveal that the majority

line 391: on the causes of body weight changes in animals infected with trypanosomes, especially after

line 413: due to the mutation or absence of a drug uptake gene

line 419: In summary, this study has found that there is variation in

7. PLOS authors have the option to publish the peer review history of their article (what does this mean?). If published, this will include your full peer review and any attached files.

Reviewer #1: No

Reviewer #2: Yes: Jay Taylor

---

## [Author Response · Author response to Decision Letter 1]

2 Jul 2020

line 91: The authors remark that the mouse has a "genetic similarity to humans estimated to be 97.5%",but I don't know any way of defining genetic similarity that would give such a high number. According to the website https://www.genome.gov/10001345/importance-of-mouse-genome, the average similarity between the shared protein-coding genes is around 85%. Please delete or modify this claim. We thank the reviewer for this observation and has modified this claim as suggested. It now reads ‘’……. genetic similarity to humans estimated to be 85 %,……..’’

Margin line 95 of the marked manuscript

line 45: A cure rate of at least 80% was achieved for all test isolates This has now been rephrased to read as suggested. Margin lines 45-46 of the marked manuscript

line 48: evidence of variation in ... confirms that this variation is We have corrected this as suggested. Margin line 50 of the marked manuscript.

line 73: in different foci differ in both their Corrected as suggested. Margin line 74 –marked manuscript

line 87: this variation Corrected as suggested. Margin line 88-marked manuscript.

line 143: once a week using a Corrected as suggested. Margin lines 144-145- marked manuscript

line 146: The classification of trypanosome virulence Corrected as suggested. Margin line 147-marked manuscript

line 165: were prepared Corrected. Margin line 166

line 179: body weight changes Corrected. Margin line 180

line 181: The general linear model in SAS was ... between the means of the four virulence classes. Corrected as suggested. Margin lines 182-183

line 190: exhibited variation in Corrected. Margin line 191

line 196: Mice infected with the acute clones had MST ranging from 18.1 +- 0.7 (KETRI 2487) to 26 +- 2.0 for... (You don't need to keep repeating the phrase mean +- SEM for each set of numbers.) In agreement with the reviewer and has deleted the repeated phrase mean +- SEM. Margin lines 197-198

line 207: The overall MST was 8.7 +- for the very acute clones, 21.6 +- 1.0 for the acute clones, Corrected as suggested. Margin lines 207-209 –marked manuscript.

line 227: are shown in Table and Figure 2(v). Corrected .margin line 229-marked manuscript

line 231: clone specific variation was observed Corrected as suggested. Margin line 233

line 257: The pre-infection PCV data for all infected Corrected. Margin 259

line 258: The PCV of the infected mice declined significantly following infection when compared Corrected as suggested. Margin line 260-261-marked manuscript.

line 308: at least 80% cure rates for all drug dose regimens Corrected. Margin line 310-marked manuscript

line 309: a few cases of relapse Corrected. Margin line 311-marked manuscript.

line 314: without detectable parasites. Corrected. Margin 316- marked manuscript

line 343: a previous report of variation in virulence Corrected. Margin line 345-marked manuscript

line 356: differences in isolate virulence are an intrinsic attribute Corrected. margin 358- marked manuscript

line 357: Parasitemia progression differed significantly among Tbr isolates assigned to different virulence classes on the basis of survival time Corrected as suggested. Margin line 359-360-marked manuscript

line 363: acute infections result from Corrected Margin line 366- marked manuscript.

line 371: they reveal that the majority Corrected. Margin line 374-marked manuscript.

line 391: on the causes of body weight changes in animals infected with trypanosomes, especially after Corrected. Margin line 393-394-marked manuscript

line 413: due to the mutation or absence of a drug uptake gene Corrected. Margin line 416

line 419: In summary, this study has found that there is variation in Corrected. Margin line 422-marked manuscript.

---

## [Editor Report · Decision Letter 2]

7 Jul 2020

Differential virulence of Trypanosoma brucei rhodesiense isolates does not influence the outcome of treatment with anti-trypanosomal drugs in the mouse model

PONE-D-20-02583R2

Dear Dr. Ndungu,

We’re pleased to inform you that your manuscript has been judged scientifically suitable for publication and will be formally accepted for publication once it meets all outstanding technical requirements.

Kind regards,

Martin Chtolongo Simuunza, PhD

Academic Editor

PLOS ONE
---

## [Editor Report · Acceptance letter]

16 Jul 2020

PONE-D-20-02583R2 

Differential virulence of *Trypanosoma brucei rhodesiense* isolates does not influence the outcome of treatment with anti-trypanosomal drugs in the mouse model 

Dear Dr. Ndung’u:

I'm pleased to inform you that your manuscript has been deemed suitable for publication in PLOS ONE. Congratulations! Your manuscript is now with our production department. 

Kind regards, 

on behalf of

Dr. Martin Chtolongo Simuunza 

Academic Editor

PLOS ONE